# Privacy-Aware Machine Unlearning with SISA for Reinforcement Learning–Based Ransomware Detection

## ABSTRACT

Ransomware detection systems increasingly rely on behavior-based machine learning to address evolving attack strategies. However, emerging privacy compliance, data governance, and responsible AI deployment demand not only accurate detection but also the ability to efficiently remove the influence of specific training samples without retraining the models from scratch. In this paper, we present a privacy-aware machine unlearning evaluation framework for RL-based ransomware detection built on Sharded, Isolated, Sliced, and Aggregated (SISA) training. The framework enables efficient data deletion by retraining only the affected model shards rather than the entire detector, substantially reducing the retraining cost while preserving the detection performance. We conducted a controlled comparative study using value-based RL agents such as Deep Q-Network (DQN) and Double Deep Q-Network (DDQN), under identical experimental settings, including a cost-sensitive reward design and 5-fold stratified cross-validation on Windows 11 behavioral ransomware telemetry. The detection confidence was evaluated using a continuous Q-score margin, enabling ROC–AUC analysis beyond binary predictions. For unlearning, the dataset was partitioned into five shards with majority-vote aggregation, and a fast-unlearning path was evaluated by deleting 5% of the samples from a single shard and retraining only that shard. The experimental results show that SISA-based unlearning incurs negligible utility degradation (≤0.2% absolute F1) while achieving substantial retraining time reduction compared with full SISA retraining. The DDQN exhibits slightly improved stability and lower utility loss relative to the DQN, although both agents maintain near-identical in-distribution performance after unlearning. These findings demonstrate that SISA provides a practical, auditable, and computationally efficient unlearning mechanism for RL-based ransomware detection, thereby supporting privacy-aware deployment without compromising security effectiveness.

## CCS CONCEPTS

• Security and privacy → Ransomware detection • Computing methodologies → Reinforcement learning • Security and privacy → Privacy-preserving systems

## KEYWORDS

Ransomware detection, machine unlearning, reinforcement learning, DQN, DDQN, responsible AI.

## 1 Introduction

The exponential growth of machine learning (ML) and artificial intelligence (AI) in cybersecurity has fundamentally transformed the way organizations detect and respond to cyber threats [1]. Ransomware has emerged as one of the most destructive and rapidly evolving threats in the digital landscape, with attacks increasing in both frequency and sophistication. Ransomware detection systems increasingly rely on machine learning to cope with evolving attack strategies; however, they face mounting pressures from privacy regulations, data governance requirements, and responsible AI deployment mandates [2]. While achieving high detection accuracy is critical for security, modern systems must also address an often-overlooked challenge: the ability to efficiently remove the influence of specific training samples without retraining the models from scratch. This need arises from privacy compliance (e.g., GDPR "right to be forgotten"), data correction requests, and ethical deployment practices [3].

Machine Unlearning has emerged as a cutting-edge technique designed to address this critical gap by enabling the selective removal of training data influence from trained models without complete retraining [4]. Traditional machine unlearning approaches require full model retraining, making them computationally prohibitive for use in continuous security systems. The sharded, isolated, sliced, and aggregated (SISA) training framework represents a particularly promising approach to machine unlearning. By strategically partitioning the training data into independent shards during the initial training phase and training separate constituent models on each shard, SISA enables efficient unlearning through the retraining of only the affected shard from its last valid checkpoint [5].

While unlearning has been studied in supervised learning [6], [7], its integration with reinforcement learning especially in security-sensitive domains remains underexplored. To address this gap, this study proposes an integrated framework for privacy-aware machine unlearning specifically designed for RL-based ransomware detection systems. Our approach extends the SISA framework to RL-based ransomware detection without degrading detection utility or incurring prohibitive retraining costs while maintaining the dynamic threat detection capabilities essential for modern cybersecurity. We adopted Sharded, Isolated, Sliced, and Aggregated (SISA) training to enable

efficient shard-level retraining and evaluated its interaction with value-based RL agents (DQN and DDQN) under identical experimental settings. Using Windows 11 behavioral telemetry and a Q-score–based ROC evaluation, we quantified the detection performance, utility preservation, and computational overhead before and after one-shard unlearning. This study provides an empirical foundation for deploying RL-based ransomware detectors in privacy-constrained environments. This study makes four key contributions.

1. We present the first systematic study of SISA-based machine unlearning applied to RL-based ransomware detection, enabling the efficient unlearning of sensitive training samples without compromising the detection performance.

2. We evaluated the DQN and DDQN under identical reward designs, hyperparameters, and cross-validation settings to isolate the algorithmic effects.

3. We demonstrate efficient one-shard unlearning that achieves negligible utility degradation (0.2% absolute F1) while reducing the retraining cost to near-baseline levels.

4. We provide an auditable and deployment-oriented assessment of unlearning behavior that is suitable for responsible AI requirements in security systems.

The remainder of this paper is organized as follows. Section 2 reviews the related work on behavior-based ransomware detection, RL in ransomware analytics, and machine unlearning with SISA. Section 3 describes the complete methodology, such as dataset secription, the RL formulation, Q-score evaluation, and SISA-based unlearning protocol. Section 4 details the experimental setup, including the architecture of the Q-network and the hyperparameters used for implementation. Section 5 presents the results. Section 6 discusses the implications, limitations, and directions for verifiable and large-scale unlearning. Finally, Section 7 concludes the study.

## 2. Related Work

### Ransomware Detection via Behavioral Analysis
Behavior-based ransomware detection has emerged as a robust alternative to static signature methods, which fail in the presence of obfuscation and polymorphism [8]. Early dynamic analysis studies have demonstrated that API calls, file system activity, and registry behavior are effective discriminators for ransomware [9], [10], [11]. Subsequent studies have adopted machine learning classifiers and deep neural networks over behavioral traces to improve detection accuracy [12], [8], [11], [13]. Although these approaches achieve high in-distribution performance, they assume immutable training data and do not address post-deployment data removal or privacy constraints.

### Reinforcement Learning for Cybersecurity and Ransomware
Recent research has explored reinforcement learning for malware and ransomware detection, motivated by its ability to encode asymmetric security costs and adapt to evolving behavior [14]. Value-based methods, such as DQN and DDQN, have shown promise for structured behavioral telemetry owing to their stability and sample efficiency. DQN has been applied to malware detection and response optimization, whereas Double DQN (DDQN) [15] improves stability by mitigating Q-value overestimation. However, existing RL-based ransomware detectors focus exclusively on detection accuracy [16] or defense mechanisms [17]. The implications of data deletion or unlearning on learned reinforcement learning (RL) policies remain largely unexplored.

### Machine Unlearning and SISA
Machine unlearning addresses regulatory requirements such as the GDPR right to erasure by enabling models to forget specific training samples [3], [18]. Among scalable approaches, SISA (Sharded, Isolated, Sliced, and Aggregated) training allows efficient unlearning by retraining only the affected data shards [5], [19]. Prior studies have validated SISA primarily in supervised learning settings and evaluated its computational efficiency and privacy guarantees [6], [7]. Its application to reinforcement learning–based security systems, particularly ransomware detection, has received little attention

### Gap Addressed by This Study
To the best of our knowledge, no prior study has systematically evaluated SISA-based unlearning in reinforcement-learning-based ransomware detection. Existing studies have focused on RL detection without unlearning or on unlearning mechanisms without considering RL-specific evaluation, cost-sensitive rewards, or security-critical constraints. This study bridges these gaps by evaluating DQN and DDQN under identical settings and quantifying the impact of one-shard SISA unlearning on the detection utility and runtime. This evaluation addresses the gap between adaptive ransomware detection and responsible artificial intelligence (AI) deployment.

## 3. Methodology
This section presents a privacy-aware reinforcement learning framework for ransomware detection using SISA-based machine unlearning. The proposed methodology supports efficient data deletion through sharded retraining, in line with responsible AI requirements.

### 3.1 Dataset and Feature Representation

Experiments were conducted on our custom behavioral ransomware dataset comprising 2000 executables (1000 ransomware and 1000 benign). Ransomware samples span 30 high-impact families selected based on multi-year (2019–2024) threat-intelligence prevalence and multi-vendor verification, sourced from MalwareBazaar and VirusShare, while benign samples were collected from trusted repositories (SnapFiles, PortableApps, GitHub). Individual ransomware samples were retained only if ≥45 VirusTotal engines flagged them as malicious, ≥15 explicitly labeled them as ransomware, and ≥10 engines agreed on family attribution.

Samples were executed in the ANY.RUN Windows 11 sandbox to ensure high-fidelity behavioral capture and avoid legacy Cuckoo limitations, producing JSON reports with ~11k raw indices and ~250 behavioral fields. To ensure reproducibility and prevent data leakage, preprocessing steps, including biased entry removal, categorial encoding, feature scaling, and standardization, were performed to create an optimal dataset with selected ransomware features covering the filesystem, registry, process, API, network, CryptoAPI, incident rules, reputation scores, and multi-process behaviors. The final output was a 103-dimensional feature vector suitable for ML- and RL-based ransomware detection.

### 3.2 Problem Formulation

Let the labeled behavioral dataset be

$$\mathcal{D} = \{(\mathcal{X}_i, \mathcal{Y}_i)\}_{i=1}^{N}, \qquad \mathcal{Y}_i \in \{0,1\} \tag{1}$$

Where $N=2000$, $\mathcal{Y}_i = 1$ denotes ransomware and $\mathcal{Y}_i = 0$ denotes benign behavior.

The objective is to learn the detection function:

$$f(\mathcal{X}) \rightarrow \{0,1\} \tag{2}$$

that maximizes the detection accuracy while enabling post-training deletion of selected samples without full retraining.

### 3.3 Reinforcement Learning Environment Design

Ransomware detection is modelled as a binary decision-making problem using reinforcement learning, where each training instance corresponds to a state, an action, and a reward transition.

**State**

Each behavioural feature vector corresponds to a state,

$$S \in \mathbb{R}^d \tag{3}$$

where d= number of total features=103

**Action**

The agent (Q-network) learns to classify behavioral states based on two discrete actions:

$$a \in \{0,1\} \tag{4}$$

where 0 denotes benign and 1 denotes ransomware, respectively.

**Reward Function (Cost-Sensitive)**

To reflect the asymmetric risk in ransomware detection, all experiments used a fixed cost-sensitive reward function that penalized false negatives more severely than false positives. The reward function used is defined as

$$r(\mathcal{Y}, a) = \begin{cases} +1, & a = \mathcal{Y} \\ -2, & \mathcal{Y} = 1 \wedge a = 0 \quad (false\ negative) \\ -0.5, & \mathcal{Y} = 0 \wedge a = 1 \end{cases} \tag{5}$$

where *y* denotes the ground-truth label. This reward configuration reflects the standard security practice, where undetected ransomware poses a substantially greater risk than benign misclassification.

### 3.4 Value-Based Deep Reinforcement Learning Algorithms

This study focuses on value-based reinforcement learning methods because of their stable optimization behavior and explicit state–action value estimation when applied to structured behavioral telemetry. In addition, value-based methods provide action–value margins, which are exploited for confidence ranking and Q-score–based ROC analysis under privacy-driven deletion constraints.

Two value-based deep reinforcement learning algorithms were evaluated:

- Deep Q-Network (DQN)
- Double Deep Q-Network (DDQN)

The DQN and DDQN shared identical experimental settings, including the network architecture, replay buffer size, learning rate, exploration schedule, discount factor, training steps, and random seeds. The only difference lies in the temporal difference (TD) target computation. This ensures that performance differences can be attributed to the value estimation mechanism rather than confounding hyperparameters.

Both DQN and DDQN learn an action–value function $Q_\theta(s, a)$ parameterized by a neural network with weights $\theta$. Learning proceeds by minimizing the TD loss over transitions sampled from an experience replay buffer. Specifically, the optimization objective is

$$\mathcal{L}(\theta) = \mathbb{E}_{(s,a,r,\acute{s}) \sim \mathcal{D}}[(\mathcal{Y} - Q_\theta(s, a)^2] \tag{6}$$

where $D$ denotes the replay buffer and $y$ is a target value computed using a separate target network. The discount factor $\gamma \in (0,1)$ controls the weighting of future rewards.

In the DQN, the TD target is computed by directly maximizing the target network's estimated action value at the next state:

$$\mathcal{Y}^{DQN} = r + \gamma max_{\acute{a}}Q_{\theta} - (\acute{s}, \acute{a}) \qquad (7)$$

Although effective, this formulation is known to suffer from overestimation bias, as the same value function is used for both action selection and evaluation.

DDQN mitigates the overestimation bias by decoupling action selection from action evaluation. The online network selects the next action, whereas the target network evaluates its value as follows:

$$\mathcal{Y}^{DDQN} = r + \gamma Q_{\theta} - (\acute{s}, argmax_{\acute{a}}Q_{\theta}(\acute{s}, \acute{a})) \qquad (8)$$

Here, $Q_{\theta}-$ denotes the frozen target network used for value evaluation, and $Q_{\theta}$ represents the online network used for action selection. This modification improves stability without increasing computational complexity, making the DDQN particularly suitable for structured behavioral telemetry.

### 3.5 Cross-Validation and Training Protocol
All experiments were conducted using 5-fold stratified cross-validation with a fixed random seed to ensure that performance differences arose solely from algorithmic behavior rather than configuration bias. For each fold:
- the RL agent is trained on 80% of the data,
- evaluated on the held-out 20%,
- The results were aggregated across the folds.

Out-of-fold (OOF) predictions were used to compute the confusion matrices, ROC curves, and summary statistics.

### 3.6 Q-Score–Based ROC Evaluation
Reinforcement learning agents do not generate calibrated probabilities. To enable ROC analysis, we derived a continuous confidence score from the learned action-value function, called the Q-score. For each sample x, the Q-score is defined as

$$Q - score(\mathcal{X}) = Q_{\theta}(\mathcal{X}, a = 1) - Q_{\theta}(\mathcal{X}, a = 0) \qquad (9)$$

The margin indicates confidence in predicting ransomware or benign behavior. ROC curves and AUC values used Q-scores rather than discrete actions, providing a faithful assessment of ranking performance and confidence.

### 3.7 SISA-Based Unlearning with Ensemble Aggregation

Machine unlearning aims to remove the influence of specific training samples from deployed models in response to privacy regulations, data correction requirements, or security policies. In learning-based security systems, full retraining after every deletion request is often computationally infeasible, particularly under continuous operational constraints.

To enable efficient unlearning, we adopt a Sharded, Isolated, Sliced, and Aggregated (SISA) training paradigm. As illustrated in Figure 1, SISA partitions the training dataset into multiple disjoint shards, trains independent models on each shard in isolation, and aggregates the predictions at the inference time. This design ensures that the influence of individual samples is confined to a limited subset of models, thereby enabling targeted retraining.

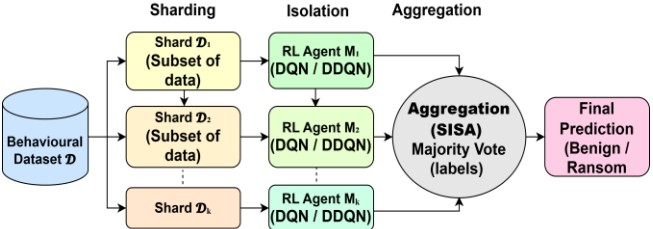

**Figure 1.** Workflow of the SISA-based training and aggregation framework.

In dataset partitioning (sharding phase), the training dataset is partitioned into $M$=5 disjoint shards, and an independent reinforcement learning agent (DQN or DDQN) is trained on each shard. Each shard-level agent learns an action–value function using only its local subset of data, thereby ensuring isolation between the shards.

At inference time, each agent produces a class prediction based on its learned Q-function as follows:

$$\acute{\mathcal{Y}}_m(\mathcal{X}) = argmax_a Q_m(\mathcal{X}, a) \qquad (10)$$

where $Q_m$ denotes the action–value function learned by the $m$-th shard agent.

These shard-level predictions correspond to the isolation stage shown in Figure 1, where each RL agent independently evaluates the input sample. The final classification is obtained via majority voting across all shard-level predictions as follows:

$$\hat{\mathcal{Y}}SISA(\mathcal{X}) = \mathbb{I}[\frac{1}{M}\sum_{m=1}^{M}\acute{\mathcal{Y}}_m(\mathcal{X}) \geq 0.5] \qquad (11)$$

where $\mathbb{I}[.]$ denotes the indicator function. This ensemble aggregation improves robustness while preserving shard isolation.

When a deletion request is issued, only the shard(s) containing the affected samples are retrained, whereas all

other shard models remain unchanged. In this study, we evaluate a fast-unlearning path in which 5% of samples from a single shard are removed and only that shard is retrained. This approach substantially reduces the unlearning cost while preserving the predictive utility.

Overall, the SISA framework balances computational efficiency and deletion fidelity, making it suitable for privacy-aware ransomware detection systems.

## 3.8 Utility Preservation Measurement

To quantify the impact of privacy-driven unlearning on the model performance, we measured the utility preservation using the change in the F1-score before and after unlearning. The utility drop is defined as

$$\Delta F1 = |F1_{before} - F1_{after}| \tag{12}$$

This absolute measure provides a non-negative and symmetric estimate of performance change, where smaller $\Delta F1$ indicates stronger utility preservation.

Algorithm 1 summarizes the evaluation pipeline of the value-based reinforcement learning framework, detailing the training, Q-score detection, SISA-based unlearning, and utility preservation under identical settings.

---

**Algorithm 1: Privacy-Aware Evaluation of Value-Based Reinforcement Learning with SISA Fast Unlearning**

**Input:** Behavioral dataset $D = \{(x_i, y_i)\}_{i=1}^{N}$ with 103-dimensional feature vectors; algorithms {*DQN, DDQN*}; folds *K*=5; time steps *T;* discount factor *γ*; Reward *R2.*; SISA shards *M;* and forgot fraction *f.*

**Output:** Cross-validated ID performance, Q-score ROC-AUC, runtime cost, and utility preservation before and after unlearning.

```
1:  Split dataset D into K stratified folds
2:  for each algorithm A ∈ {DQN, DDQN} do
3:  |   Initialize OOF_true, OOF_pred, OOF_qscore ← ∅
4:  |   for fold k = 1 to K do
5:  |   |   (D_tr, D_te) ← StratifiedSplit(D, k)
6:  |   |   (X_tr, X_te) ← Standardize(D_tr.X, D_te.X)
7:  |   |   ▷ Base RL training
8:  |   |   Initialize Q_online, Q_target, replay buffer B
9:  |   |   for t = 1 to T do
10: |   |   |   s ← X_tr[t mod |X_tr|]
11: |   |   |   s′ ← X_tr[(t+1) mod |X_tr|]
12: |   |   |   a ← ε-greedy(Q_online, s)
13: |   |   |   r ← R2(y(s), a)
14: |   |   |   Store (s, a, r, s′) in B
15: |   |   |   if |B| ≥ batch_size then
16: |   |   |   |   if A = DDQN then
17: |   |   |   |   |   a* ← argmax_a Q_online(s′, a)
18: |   |   |   |   |   y_TD ← r + γ · Q_target(s′, a*)
19: |   |   |   |   else
20: |   |   |   |   |   y_TD ← r + γ · max_a Q_target(s′, a)
21: |   |   |   |   end if
22: |   |   |   |   Update Q_online using y_TD
23: |   |   |   end if
24: |   |   |   Periodically update Q_target
25: |   |   end for
26: |   |   ▷ In-distribution evaluation
27: |   |   ŷ ← argmax Q_online(X_te)
28: |   |   q ← Q_online(X_te,1) − Q_online(X_te,0)
29: |   |   Append y_te, ŷ, q to OOF sets
30: |   |   ▷ SISA training
31: |   |   Partition D_tr into M shards
32: |   |   Train one RL agent per shard
33: |   |   ŷ_SISA_before ← MajorityVote(shard_models, X_te)
34: |   |   F1_before ← F1(y_te, ŷ_SISA_before)
35: |   |   ▷ Fast unlearning (one shard)
36: |   |   Select shard m*
37: |   |   Select deletion set F ⊂ shard m* with |F| = ⌈ f · |shard m*| ⌉ and remove F
38: |   |   Retrain only shard m* on retained data
39: |   |   ŷ_SISA_after ← MajorityVote(updated_models, X_te)
40: |   |   F1_after ← F1(y_te, ŷ_SISA_after)
41: |   |   UtilityDrop ← |F1_before − F1_after|
42: |   |   Record runtime and utility metrics
43: |   end for
44: |   Aggregate fold-wise results
45: |   Compute OOF confusion matrix and Q-score ROC–AUC
46: end for
```

---

## 4. Experimental Setup

All experiments were conducted in a CPU-only Google Colab Pro environment using an Intel Xeon-class virtual machine with 51 GB of RAM, ensuring that the proposed framework remains practical for real-world deployment without GPU dependence. The implementation was developed in Python 3.10, using PyTorch 2.1 and scikit-learn 1.3. This setup reflects realistic enterprise constraints and avoids reliance on specialized hardware-accelerators. The architectural design and training hyperparameters are presented in Tables 1 and 2, respectively.

**Table 1.** Q-Network Architecture

| Layer | Configuration |
|---|---|
| Input | 103 neurons (behavioral feature vector) |
| Hidden Layer 1 | 128 neurons, ReLU activation |
| Hidden Layer 2 | 128 neurons, ReLU activation |
| Output | 2 neurons (Q-values for class 0: benign, class 1: ransomware) |

**Table 2**. Training Hyperparameters

| Hyperparameter | Value |
|---|---|
| Optimizer | Adam, learning rate α = 0.001 |
| Batch Size | 64 |
| Training Budget | 10,000 timesteps per agent (streaming updates) |
| Discount Factor (γ) | 0.1 |
| Replay Buffer Size | 50,000 |
| Target Network Update Frequency | Every 500 training steps |
| Exploration Strategy | ε-greedy, ε = 1.0 → 0.05 (linear decay over 5,000 steps) |
| Loss Function | Smooth L1 (Huber) loss |
| Reward Function | Cost-sensitive (R2): +1 (correct), −2 (false negative), −0.5 (false positive) |

## 5. Results

This section reports the experimental results of the proposed privacy-aware reinforcement learning framework with SISA-based machine unlearning. The evaluation focuses on (i) baseline detection performance, (ii) utility preservation before and after one-shard SISA unlearning, and (iii) computational efficiency and retraining overhead.

### 5.1 Baseline Detection Performance

This subsection presents the in-distribution (ID) ransomware detection performance of the baseline value-based reinforcement learning agents before any unlearning is applied. Performance is reported using the F1-score, worst-fold F1, and Q-score–based ROC–AUC, alongside the average training and inference time per fold.

Table 3 summarizes the 5-fold cross-validated in-distribution ransomware detection performance of baseline value-based reinforcement learning agents. Both DQN and DDQN achieved near-perfect detection accuracy, with mean F1-scores above 0.99 and Q-score ROC–AUC values exceeding 0.998, demonstrating a strong discriminative capability for behavioral telemetry. The DDQN marginally outperformed the DQN in terms of the mean F1-score and exhibits improved worst-fold stability, consistent with its reduced overestimation bias in value estimation.

**Table 3.** In-Distribution Detection Performance (Baseline RL Agents)

| Model | ID F1 (mean ± std) | Worst-Fold ID F1 | OOF AUC (Q-score) | Train Time (s) | Inference Time (s) |
|-------|------|------|------|------|------|
| DQN | 0.9920 ± 0.0045 | 0.9850 | 0.9987 | 23.14 | 0.0365 |
| DDQN | 0.9925 ± 0.0025 | 0.9900 | 0.9983 | 25.13 | 0.0457 |

Figure 2 further analyzes the winning model (DDQN) using an out-of-fold confusion matrix. The model correctly classified 99.3% of benign samples and 99.2% of ransomware samples, with false-positive and false-negative rates limited to 0.7% and 0.8%, respectively. This balanced error profile confirms that the cost-sensitive reward formulation effectively prioritizes ransomware detection while maintaining low benign misclassification rates.

Figure 3 shows the ROC curve derived from the proposed Q-score margins. The DDQN achieved an OOF ROC–AUC of approximately 0.998, indicating a near-perfect ranking performance. By leveraging action-value differences rather than calibrated probabilities, this

evaluation provides a faithful assessment of confidence and decision robustness for reinforcement learning–based detectors. Overall, these results establish the DDQN as a stable and reliable baseline, forming a strong reference point for subsequent privacy-driven SISA unlearning analyses.

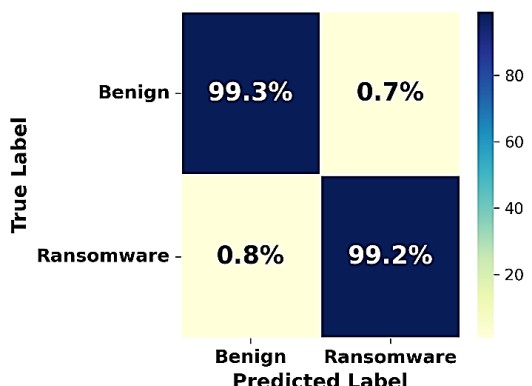

**Figure 2.** Confusion Matrix for DDQN Baseline Detection.

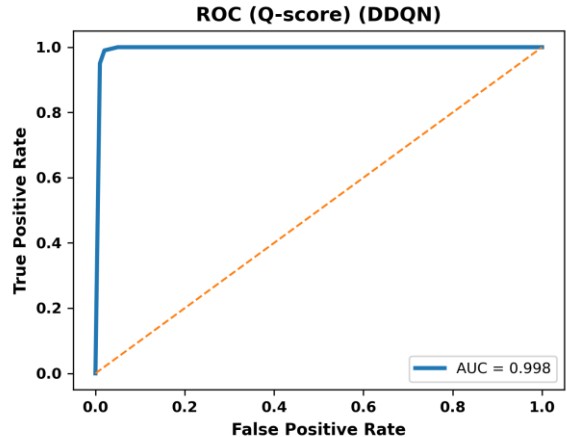

**Figure 3.** Q-Score ROC Curve for DDQN Baseline Detection

### 5.2 Privacy-Driven Utility Preservation under SISA Unlearning

To evaluate the impact of privacy-driven unlearning, we apply SISA-based sharded retraining with M = 5 shards, followed by a fast-unlearning path, in which 5% of samples are removed from a single shard, and only the affected shard is retrained. Table 4 reports the detection performance before and after unlearning, which was measured using ensemble majority-vote predictions on the test set.

**Table 4.** Utility Preservation Before and After One-Shard SISA Unlearning

| Model | SISA F1 (Before) | SISA F1 (After) | Utility Drop (ΔF1) |
|-------|------|------|------|
| DQN | 0.9787 | 0.9782 | 0.0005 |
| DDQN | 0.9806 | 0.9806 | 0.0000 |

Across both DQN and DDQN, the observed utility degradation was negligible. The average F1-score drop was approximately 0.05% for DQN and 0.00% for DDQN, indicating that one-shard unlearning preserves almost all detection capabilities.

These results confirm that SISA-based unlearning enables practical data deletion with a minimal impact on ransomware detection effectiveness, satisfying privacy requirements without compromising operational security.

### 5.3 Computational Efficiency and Retraining Overhead

Table 5 summarizes the average computational costs across five folds for both the DQN and DDQN. Full SISA training increases the cost by approximately fivefold relative to the baseline training, reflecting the overhead of maintaining shard isolation. However, the one-shard unlearning strategy retrains only the affected shard, reducing the unlearning time to near-baseline levels ($\approx$22–24 s). This confirms that SISA-based unlearning enables efficient, scalable, and privacy-compliant model updates without incurring the prohibitive costs of full retraining.

**Table 5.** Computational Cost Comparison: Baseline Training vs. SISA Unlearning

| Model | Baseline Train (s) | SISA Full Train (s) | One-Shard Unlearn (s) |
|-------|--------------------|--------------------|-----------------------|
| DQN   | 23.14              | 113.30             | 22.40                 |
| DDQN  | 25.13              | 123.21             | 23.98                 |

Figure 4 illustrates the fold-wise runtime breakdown for the DDQN model, comparing the baseline single-agent training, full SISA training with five shard-level agents, and one-shard retraining for privacy-driven unlearning purposes. Baseline training remains stable across folds ($\approx$25 s), whereas full SISA training incurs a higher cost owing to independent shard-wise model replication. In contrast, one-shard retraining consistently requires a runtime comparable to that of baseline training, demonstrating the effectiveness of the proposed fast-unlearning path.

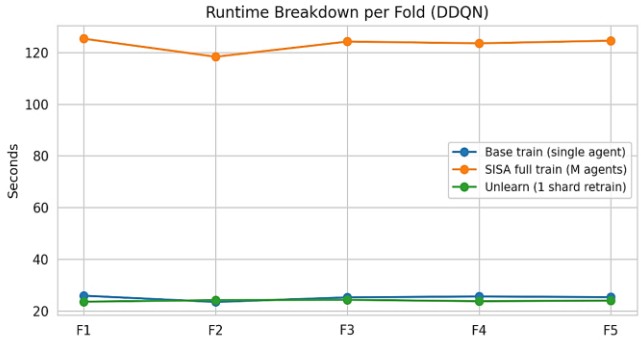

**Figure 4**. Runtime Breakdown per Fold (DDQN).

This reduction in retraining overhead enables continuous operation while supporting privacy-driven deletion, essential for real-world ransomware detection systems subject to regulatory and operational constraints.

### 6. Discussion

This study examines whether privacy-aware machine unlearning can be incorporated into reinforcement learning–based ransomware detection without significantly affecting the detection performance. The results show that the SISA-enabled DDQN model preserves near-baseline in-distribution accuracy while enabling efficient shard-level deletion with substantially lower retraining costs than full-model retraining. This suggests that unlearning can be integrated into security-focused RL systems with limited operational disruptions.

The findings are consistent with earlier work demonstrating the effectiveness of value-based reinforcement learning for behavioral malware detection under cost-sensitive reward structures. However, most existing approaches assume static training data and do not address the post-training deletion requirements. By introducing shard-isolated retraining, this study extends prior methods to scenarios in which data removal is required after deployment. The observed stability of the DDQN further indicates that reduced value overestimation may help preserve the model behavior following partial retraining.

This study focuses on value-based reinforcement learning rather than policy-gradient or actor–critic methods because the evaluation relies on explicit state–action value estimates. In particular, the Q-score margin $Q(\mathcal{X}, 1) - Q(\mathcal{X}, 0)$ is used for confidence ranking and ROC analysis, which cannot be directly obtained from stochastic policy outputs. Therefore, value-based methods provide interpretable decision margins under privacy-driven deletion constraints, making them well-suited for the security-oriented evaluation setting considered in this study.

Although the evaluation is limited to single-shard deletions and does not include formal verification of forgetting, the results indicate that practical, privacy-aware unlearning can be integrated into security systems with minimal operational impact. This study represents an initial step toward the responsible deployment of RL-based ransomware detection and motivates future studies on verifiable and large-scale unlearning.

### 7. Conclusion and Future Work

This study presents a privacy-aware reinforcement learning framework for ransomware detection that integrates SISA-based machine unlearning into value-based RL agents. Through a controlled comparative evaluation of DQN and

DDQN under identical experimental settings on Windows 11 behavioral ransomware telemetry, we demonstrated that one-shard SISA unlearning enables efficient data deletion with a negligible impact on detection performance, while substantially reducing retraining overhead compared to full-shard retraining. The results confirm that DDQN provides stable and robust behavioral detection, achieving near-perfect in-distribution performance (F1 > 0.99, AUC > 0.998) and that privacy-driven unlearning can be achieved without compromising baseline security guarantees. Our findings establish SISA as a practical, auditable, and computationally efficient unlearning mechanism for RL-based ransomware detection, thereby supporting responsible AI deployment in security-critical applications. This study represents an initial step toward bridging the gap between privacy compliance requirements and operational security effectiveness, demonstrating that emerging responsible AI standards can be integrated into production security systems with minimal disruption.

Future research will extend SISA-based unlearning to encompass multi-shard and sequential deletion scenarios, targeted sensitive-sample removal, and explore unlearning within broader datasets and adversarial contexts. In addition, we will investigate policy-level and actor–critic methods under SISA-based unlearning as a future research direction. Furthermore, we will examine verifiable unlearning mechanisms, including oracle-based forgetting tests and membership inference to provide more robust evidence that the model has genuinely forgotten the deleted data and to enhance the credibility of the privacy guarantees.

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
