# OpenReview forum: "Privacy-Aware Machine Unlearning with SISA for Reinforcement Learning–Based Ransomware Detection"
_ACM.org/TheWebConf/2026/Workshop/TIME — TIME 2026 Poster_

### Official Review · Reviewer_zCyW · 2025-12-30
**A Rigorous and Well-Motivated Framework for Privacy-Aware Reinforcement Learning–Based Ransomware Detection**

**Rating:** 7
**Confidence:** 3

**Review:**

This paper presents a comprehensive and technically sound study on privacy-aware machine unlearning for reinforcement learning–based ransomware detection. The authors propose a structured framework that integrates SISA-based unlearning with value-based reinforcement learning (DQN and DDQN), enabling efficient post hoc data deletion while preserving detection performance. The work is well motivated by practical regulatory and deployment constraints and addresses an important gap at the intersection of security, machine learning, and responsible AI.

A major strength of the paper lies in its methodological rigor and clarity. The problem formulation is precise, the reinforcement learning setup is well-defined, and the use of cost-sensitive rewards appropriately reflects real-world ransomware risk asymmetries. The experimental protocol is carefully controlled, with consistent hyperparameters, cross-validation, and clearly defined evaluation metrics. The use of Q-score–based ROC analysis is particularly well-justified for reinforcement learning settings where probability calibration is nontrivial.

The integration of SISA-based unlearning into an RL pipeline is novel in this context and convincingly demonstrated. The experimental results show that one-shard unlearning achieves near-zero degradation in detection performance while significantly reducing retraining cost. The quantitative analysis, including utility preservation and runtime evaluation, provides strong empirical support for the authors’ claims. The inclusion of detailed architectural descriptions and algorithmic pseudocode further strengthens reproducibility.

The paper is also well-written and logically structured. Figures and tables are clear, and the experimental narrative is easy to follow. The discussion appropriately situates the work within existing literature on machine unlearning, reinforcement learning, and ransomware detection, while clearly articulating the practical implications of the findings.

There are, however, a few limitations worth noting. First, while the evaluation is thorough, it remains confined to a single dataset and a single family of RL algorithms. Additional experiments on alternative environments or policy-based methods could further strengthen generalizability. Second, the unlearning evaluation focuses on single-shard deletion; more adversarial or repeated deletion scenarios could further stress-test the framework. These limitations are acknowledged by the authors and do not detract significantly from the contribution.

Overall, this is a strong and well-executed paper that makes a meaningful contribution to privacy-aware machine learning and security-oriented AI systems. It is particularly well-suited for the TIME workshop audience and advances the discussion on responsible deployment of learning-based security systems.

---

### Official Review · Reviewer_8N2Z · 2026-01-03
**This paper presents a privacy-aware machine unlearning framework for reinforcement learning–based ransomware detection using SISA training, enabling efficient removal of selected training data by retraining only affected model shards. Experiments with DQN and DDQN on Windows 11 ransomware telemetry show that SISA-based unlearning achieves substantial retraining time reduction with negligible performance loss (≤0.2% F1). Overall, the framework supports practical, auditable, and privacy-compliant deployment of RL-based security systems.**

**Rating:** 5
**Confidence:** 3

**Review:**

### Strengths
1. The paper addresses machine unlearning in security-sensitive ML systems, aligning well with emerging requirements in privacy regulation, responsible AI, and data governance.
2. Applying SISA-based unlearning to RL-based ransomware detection is a meaningful extension beyond the more common supervised-learning unlearning studies.
3. The use of identical experimental settings (reward design, cross-validation, telemetry source) for DQN and DDQN strengthens the fairness and interpretability of comparisons.
4. The results convincingly show that SISA-based unlearning achieves near-zero performance loss while offering substantial computational savings.

### Weakness
1. The paper has a formatting problem.
2.The unlearning experiment removes only 5% of samples from a single shard, which may not reflect more challenging real-world deletion scenarios (e.g., multiple shards, targeted sensitive samples, or larger deletions).
3. The evaluation is limited to value-based RL (DQN/DDQN). What about policy-gradient or actor–critic methods  under SISA-based unlearning?

---

### Official Review · Reviewer_LzYv · 2026-01-05
**Privacy-Aware Reinforcement Learning for Ransomware**

**Rating:** 7
**Confidence:** 3

**Review:**

This study successfully introduces a machine unlearning mechanism into reinforcement learning–based ransomware detection, addressing both privacy regulations and practical deployment requirements. Beyond methodological novelty, the proposed approach is empirically shown to be feasible in terms of detection performance and computational efficiency.

## Strengths

- **Clear Alignment with Privacy and Regulatory Requirements**
This work directly addresses the GDPR “right to be forgotten” and responsible AI requirements by incorporating machine unlearning into the ransomware detection pipeline, rather than focusing solely on performance-driven classification. The research motivation is well-defined and highly relevant to real-world deployment.

- **First Systematic Integration of SISA with RL-Based Ransomware Detection**
While SISA has primarily been studied in supervised learning settings, this study extends it to reinforcement learning, effectively bridging a gap between RL-based security systems and machine unlearning. This integration represents a clear methodological novelty.

- **Empirical Evidence of Negligible Performance Degradation after Unlearning**
After deleting 5% of the data from a single shard, the F1-score exhibits almost no degradation (≤0.2%), while the retraining cost is substantially reduced. These results clearly demonstrate the practical effectiveness of SISA-based unlearning in this context.

## Weaknesses / Limitations

- **Evaluation Limited to Single-Shard, One-Time Deletion**
The experiments focus exclusively on a fast-unlearning scenario involving a single shard and a one-time deletion of 5% of the data. The impact of multi-shard deletions or sequential unlearning on model stability and long-term performance is not explored.

## Recommendations

- **Formatting Consistency**
It is recommended to standardize the formatting of citations throughout the paper, particularly ensuring consistent font color and style for all in-text citations, as the current presentation shows minor inconsistencies that may affect overall visual coherence.

- **Incorporate Verifiable Unlearning Evaluation Metrics**
It is recommended to integrate verification mechanisms such as membership inference, influence functions, or oracle-based forgetting tests to provide stronger evidence that the model has truly forgotten the deleted data and to enhance the credibility of the privacy guarantees.

This paper presents a well-motivated and methodologically sound study on privacy-aware reinforcement learning for ransomware detection. The proposed approach is clearly articulated and empirically validated, demonstrating effective machine unlearning with negligible performance degradation.

---

### Official Review · Reviewer_vBSi · 2026-01-07
**A strong and relevant paper for improving training data for ransomware detection algorithms**

**Rating:** 7
**Confidence:** 4

**Review:**

The paper addresses the challenge of implementing "machine unlearning" (removing specific data samples) in Reinforcement Learning-based ransomware detection systems to comply with privacy regulations like GDPR.

Quality:

A strong paper on the intersection of AI safety and Cybersecurity. Paper has good methodological rigor where they have analyzed two different algorithms DQN and DDQN to isolate algorithmic performance differences. Q-Score based ROC-AUC is also thoughtfully applied.

Results are pretty good too. SISA based fast unlearning incurs a utility loss of 0.2% absolute F1 while reducing retraining time to near-baseline levels

Clarity:

Paper is well structured, it is accessible but also has rigor. The paper clearly states the problem => methodology => Experimentation => results structure. Visuals are very helpful and the algorithm is presented correctly.

Significance:
In terms of relevance the paper is highly relevant to the venue workshop but also is overall significant to the field of cyber security and ai safety as it genuinely deals with a real problem and proposed a novel framework to solve this problem.

Final assessment:

This is a good quality, relevant and well presented paper. It should be accepted.

---

### Author Rebuttal · Authors · 2026-01-12

Consolidated Author Response to Reviewers

Submission: Privacy-Aware Machine Unlearning with SISA for Reinforcement Learning–Based Ransomware Detection

Workshop: TIME 2026 @ The Web Conference (WWW 2026)

Dear Area Chairs and Reviewers,

We sincerely thank the reviewers and Program Chairs for their thorough, constructive, and insightful feedback. We greatly appreciate your positive assessment of the novelty, technical soundness, and practical relevance of our work. We have carefully addressed all the reviewers and PC comments and revised the manuscript accordingly.

All revisions are highlighted in red in the revised manuscript, as recommended. Below, we respond to each comment point by point and explicitly indicate the corresponding changes made in the paper.


1. Response to Reviewer vBSi08

Thank you for your positive assessment of our work. We sincerely appreciate the recognition of the relevance of our paper in AI safety, cybersecurity, and privacy-aware learning. We are grateful for the acknowledgment of our DQN/DDQN comparison and Q-score-based ROC-AUC evaluation. Your observation that SISA-based fast unlearning achieves minimal utility loss while maintaining a near-baseline retraining time highlights our core contribution. In response to the feedback, the manuscript has been further refined to improve its technical clarity and presentation consistency.

2. Response to Reviewer LzYv

Thank you for your positive and constructive assessment of our work. We appreciate the recognition of the framework’s alignment with privacy and regulatory requirements, as well as the identification of methodological novelty in extending SISA-based unlearning to reinforcement learning–based ransomware detection.
Regarding the evaluation scope, the experiments focus only on single-shard, fast-path unlearning to assess correctness and computational efficiency within the scope of a conference paper. As explicitly stated in the Conclusion and Future Work section, extensions to multi-shard and sequential deletion are important directions for future research, as highlighted on page 8.Additionally, with respect to unlearning verification, we acknowledge that stronger verifiable guarantees such as oracle-based retraining comparisons, membership inference analysis, influence-based methods, and privacy leakage assessment remain important future extensions, as stated in the manuscript. We have also addressed the reviewer’s formatting suggestions by standardizing the citation style and visual consistency throughout the revised manuscript.

3. Response to Reviewer 8N2Z

Thank you for your careful evaluation and for highlighting the strengths of our work. We appreciate the recognition of the paper’s relevance to privacy regulation, responsible AI, and security-sensitive machine learning, as well as the acknowledgement of the methodological contribution of extending SISA-based unlearning to reinforcement learning–based ransomware detection under controlled experimental settings.
Regarding the unlearning scope, we acknowledge that our evaluation focuses on a single-shard, 5% deletion scenario to assess unlearning efficiency and utility preservation within the scope of this conference paper. As noted in the conclusion, extensions to multi-shard deletions, larger deletion fractions, and targeted sensitive-sample removal are future directions. With respect to the restriction to value-based reinforcement learning, we agree that extending SISA-based unlearning to policy-gradient and actor–critic methods represents an important avenue for future research and have explicitly identified this in the manuscript. We have addressed the formatting issues noted by the reviewer by standardizing the citation style, font color, and visual consistency throughout the revised manuscript.

4. Response to Reviewer zCyW

We thank you for the detailed and encouraging evaluation and sincerely appreciate the recognition of our paper's methodological rigor and the clear problem formulation. We are grateful for acknowledging the cost-sensitive reward design, experimental protocol, and Q-score ROC analysis that supports fair evaluation in reinforcement learning. We appreciate that the results show near-zero utility degradation with a substantial cost reduction.
Regarding the noted limitations, we agree that broader datasets, policy-based reinforcement learning methods, and more complex unlearning scenarios represent important future directions. These extensions are explicitly discussed in the Conclusion and Future Work sections.

5. Response to Program Chair Comments

We sincerely appreciate your conducting the AI content and reference integrity check and providing a detailed report.
Regarding the flagged references, we carefully reviewed each identified entry and corrected all reported issues. Specifically, we resolved author name mismatches, completed missing author lists, corrected paper title inconsistencies, and verified all bibliographic metadata directly against the original publication sources. We confirm that no hallucinated or unverifiable references remain in the manuscript.
All amendments have been incorporated into the revised manuscript (page 8) and are highlighted accordingly.

6. Response to Program Chair Comments

Thank you for the thorough review of the manuscript and for identifying the issues related to formatting and clarity. The suggested issues have been addressed in the revised manuscript.

• Equations (7) and (8): In the original manuscript the minus sign was not typeset clearly as a subscript of 𝜃, which led to confusion. We have revised Eqs. (7) and (8) explicitly denote the target network using the appropriate subscript notation. Specifically, 𝑄𝜃− now clearly represents the action–value function parameterized by the target network, while 𝑄𝜃 denotes the online network. This correction removes any ambiguity regarding the meaning of the negative sign and aligns the notation with the standard DQN/DDQN formulations. All corrections are reflected in the revised manuscript on page 4 and have been highlighted.

• Figure 1: We have updated the figure caption and added a concise explanation in the text to clearly describe the framework components and workflow. The updated sections are highlighted in red on pages 4–5.

• Equation (12): We reformatted the equation to a clean and unambiguous form, explicitly including the missing absolute value delimiter. The revised equation is included on page 5 and highlighted in red in the revised manuscript.

• Figure font sizes: We have standardized font sizes across all figures (Figs. 1, 2, and 4) to ensure a consistent visual appearance throughout the paper.

We believe that these revisions have significantly improved the clarity, rigor, and completeness of the manuscript while preserving its original contributions.

Once again, we sincerely thank the reviewers and Program Chairs for their valuable feedback and the opportunity to improve our work. We hope that the revised manuscript and responses satisfactorily address all of your concerns.


Sincerely,

Jannatul Ferdous and co-authors

---

### Meta-Review · Area_Chair_65sL · 2026-01-17

**Recommendation:** Accept (Poster)
**Confidence:** 3

**Metareview:**

This paper presents a privacy-aware machine unlearning evaluation framework for reinforcement learning–based ransomware detection using SISA training, enabling efficient removal of specific training data by retraining only affected model shards. Experiments with DQN and DDQN on Windows 11 behavioral telemetry show that SISA-based unlearning achieves major retraining-time savings with negligible performance loss (≤0.2% F1), demonstrating practical and auditable privacy-compliant deployment without sacrificing detection effectiveness.

Clear rebuttal has been provided to address the concerns in the review stage.

---

### Decision · Program_Chairs · 2026-01-17

Accept (Poster)